# The Effect of Acceptance and Commitment Therapy for Improving Psychological Well-Being in Parents of Individuals with Autism Spectrum Disorders: A Randomized Controlled Trial

**DOI:** 10.3390/brainsci11070880

**Published:** 2021-06-30

**Authors:** Flavia Marino, Chiara Failla, Paola Chilà, Roberta Minutoli, Alfio Puglisi, Antonino A. Arnao, Loris Pignolo, Giovambattista Presti, Francesca Pergolizzi, Paolo Moderato, Gennaro Tartarisco, Liliana Ruta, David Vagni, Antonio Cerasa, Giovanni Pioggia

**Affiliations:** 1Institute for Biomedical Research and Innovation (IRIB), National Research Council of Italy (CNR), 98164 Messina, Italy; flavia.marino@cnr.it (F.M.); chiara.failla@irib.cnr.it (C.F.); paola.chila@irib.cnr.it (P.C.); roberta.minutoli@irib.cnr.it (R.M.); alfio.puglisi@irib.cnr.it (A.P.); antoninoandrea.arnao@cnr.it (A.A.A.); gennaro.tartarisco@cnr.it (G.T.); liliana.ruta@cnr.it (L.R.); david.vagni@cnr.it (D.V.); 2Classical Linguistic Studies and Education Department, Kore University of Enna, 94100 Enna, Italy; giovambattista.presti@unikore.it (G.P.); antonio.cerasa@cnr.it (A.C.); 3S’Anna Institute, Research in Advanced Neurorehabilitation (RAN), 88900 Crotone, Italy; l.pignolo@isakr.it; 4IESCUM, European Institute for the Study of Human Behaviour, 43100 Parma, Italy; f.pergolizzi@iescum.org; 5Department of Business, Law, Economics and Consumer Affairs, IULM University, 20143 Milan, Italy; paolo.moderato@iulm.it; 6Institute for Biomedical Research and Innovation (IRIB), National Research Council of Italy (CNR), 87050 Mangone, Cosenza, Italy

**Keywords:** autism spectrum disorders, acceptance and commitment therapy, parental training, psychological well-being

## Abstract

Background: Acceptance and Commitment Therapy (ACT) has been demonstrated as effective in improving psychological well-being in several clinical domains, but there is no evidence regarding the parents of children with Autism Spectrum Disorder (ASD). Methods: In this randomized controlled trial, we evaluated the efficacy of the ACT matrix behavioral protocol in comparison to the Parent Training (PT) program, measuring several primary and secondary outcomes prior to and following treatments. Twelve parents were randomly and equally assigned to two demographically matched groups wherein individuals underwent 24 weekly meetings of ACT protocol (experimental group) or conventional PT (control group). Results: Parents enrolled in the ACT protocol demonstrated significant improvement in psychological flexibility, awareness states, personal values in everyday life, and parental stress, whereas reduced scores were elicited in parents’ perceptions of their child’s disruptive behaviors. Conclusions: The results of this randomized controlled trial, if repeated with a large number of subjects, could open the way to include ACT protocols in daily practice to support the development of new parenting skills.

## 1. Introduction

ASD is a lifelong neurodevelopmental disorder characterized by core deficits in communication skills, maladaptive behaviors, and self-regulation impairments affecting the socio-relational performance of children, as well as that of their parents [1,2]. It has been widely demonstrated that raising a child with autism involves chronic challenges consistently associated with high levels of psychological distress [3,4,5,6]. Parents often become isolated from family and friends who may not understand the child’s behavior and disability [7,8,9]. The chronic stress experienced by parents of children with ASD is reported to be greater than those experienced by parents of children with other disabilities, such as Down Syndrome, behavioral disorders, and Fragile X Syndrome [7,10,11], and is also associated with increased divorce rates [12,13].

Therefore, it is clear that family members of ASD children face three distinct challenges: understanding their children’s autism, managing their behavior, and reducing the stress caused by the behaviors themselves and social stigma [14]. Indeed, working indirectly to increase and capitalize parental resources is considered an important mediating factor in ASD intervention and may reduce maladaptive behaviors in children and increase the well-being of the whole family [15,16].

### 1.1. PT Approach

The term PT has been widely used in the ASD realm to describe a wide range of interventions [17]. Overall, the variety of PT programs could be organized into the following categories:

a. Parent support, or informative PT: a knowledge-focused approach, from which the child is an indirect beneficiary. Parent support can be further divided into (A) care support, wherein the parent receives organizational training for services coordination and access, and (B) psychoeducation to increase the understanding of the child’s needs and specificities.

b. Parent coaching: a skill-focused approach, from which the child is a direct beneficiary of the intervention. This form of PT could be further divided into (A) parent-mediated intervention for core symptoms (i.e., social communication, imitation, play), and (B) PT for maladaptive behaviors or behavioral child management (i.e., disruptive behavior, feeding, sleeping, toileting, etc.).

c. Parental psychological support: focused on ameliorating parental distress, with the child as an indirect beneficiary. This form of PT could be further separated into (A) management and sharing, focused on taking back time for themselves and sharing their emotional burden with others; and (B) acceptance-based training, focused on changing the relation between parental perspectives and acceptance of the child’s diagnosis and behavior [17,18,19].

### 1.2. Acceptance and Commitment Therapy Model

In the past few years, a new structured and systematic psychotherapy approach known as Acceptance and Commitment Therapy (ACT) has been proposed to improve parenting skills and reduce psychological distress in family members of ASD children. This represents a third wave behavioral therapy that can be useful in alleviating the psychological, emotional, and physical challenges faced by ASD parents. ACT focuses on six key processes: acceptance, cognitive defusion, being present, self-as-context, values, and committed action [20]. ACT posits that psychological distress is the result of an interaction between human language and cognition, and advocates the control of verbally constructed contingencies, rather than direct contingencies, over human behavior [20]. The goal of ACT-based therapy is to promote psychological flexibility to lead a life in line with one’s values. Psychological flexibility can be considered a behavioral repertoire that is sensitive to the presence of private events, but characterized by an adaptive, flexible, and creative response to those private events [21]. Several studies have suggested that parenting-specific psychological flexibility may be related to more adaptive parenting behaviors associated with lower levels of child problem behaviors. The ACT model has been demonstrated as useful for treating anxiety and stress [22,23], pain [24], substance use [25,26], depression [27,28], and burnout [20]. In ADHD subjects, some researchers combined this approach with a token economy to improve self-regulation skills by reducing impulsive behaviors [29]. Furthermore, the ACT intervention has also been shown to be effective in managing depression in individuals with physical disability [30]. However, there is a lack of evidence on the effectiveness of the ACT approach in reducing psychological distress and promoting well-being in parents of ASD children.

## 2. Materials and Methods

The present single-blind Randomized Controlled Trial (RCT) aims to investigate the efficacy of the ACT approach in ASD parents compared to PT for the first time. The RCT is registered at https://clinicaltrials.gov (accessed on 29 June 2021) (ID: NCT04909658) and https://www.isrctn.com/ (accessed on 29 June 2021) (ID: ISRCTN17497289). Several papers reported the beneficial effects of PT groups as classic support to increase parenting skills in managing the behavior of children with ASD [31,32,33,34] while reducing parental stress. Nevertheless, none has evaluated whether ACT may be a more powerful approach to treat psychological reactions to the stresses of caring for ASD children. We hypothesized that the psychological difficulties of parents of children with autism could decrease following the transmission of behavioral educational techniques and the promotion of psychological adjustment through cognitive defusion and acceptance strategies.

### 2.1. Inclusion Criteria

The families were recruited as part of an ongoing research program and tested at our clinical facilities. Inclusion criteria were based on children’s characteristics as follows: (1) between 4 and 10 years of age; (2) clinical diagnosis of ASD based on the DSM-5 criteria from a licensed clinical child neuropsychiatrist; (3) DSM-5 severity scores from mild (level 1) to moderate (level 2) in both social communication and restricted interests and repetitive behaviors domains; (4) a verbal and performance Griffiths Mental Development Scales, Extended Revised: 2 to 8 years (GMDS-ER 2–8) and Wechsler Intelligence Scale for Children (WISC-IV) above 70; (5) no hearing, visual, or physical disabilities that would prevent participation in the intervention; and (6) not being on psychiatric medication. All children had a previous diagnosis that was further confirmed through the assessment and the consensus of the experienced professionals on the research team (i.e., a child neuropsychiatrist and a clinical psychologist).

### 2.2. Ethics

All subjects gave informed consent for inclusion prior to their participation in the study. The study was conducted in accordance with the Declaration of Helsinki, and the protocol was approved by the Ethics Committee of the Research Ethics and Bioethics Committee (http://www.cnr.it/ethics) (accessed on 29 June 2021) of the National Research Council of Italy (CNR) (Prot. N. CNR-AMMCEN 54444/2018 01/08/2018). All of the parents of the children who participated in the study gave their consent to participate, signing a written consent form.

### 2.3. Study Design

A single-blind, randomized controlled study was conducted at the Messina unit of CNR-IRIB in Messina. The first stage was based on the recruitment of parents for the study. Next, eligible individuals underwent a clinical examination at baseline [T0]. In the third stage, participants were randomly assigned to two groups using a computer-generated randomization code. The following people were blinded to the parents’ group membership: the physicians (who carried out the clinical baseline assessment (T0) and post-treatment investigation (T1)), the primary researchers, and the data entry assistants. In the fourth stage, participants underwent ACT or PT training therapies. Treatments were conducted by expert therapists who were blinded to all clinical information and also to the aim of the study. At the end of treatment, participants from both groups were given a final evaluation (T1), using the same protocol as baseline.

### 2.4. Intervention

The two experimental protocols were developed for a total of 24 weekly meetings lasting 90 min each. The total intervention lasted six months. Parent pairs were randomly assigned to the experimental group (ACT matrix protocol) or to the control group (PT protocol) (Figure 1), using a computer-generated randomization code. Before the intervention, both groups participated in a common meeting in which the particular behavioral characteristics of children were illustrated in order to provide knowledge and awareness of the diagnosis.

Despite ACT and PT interventions differing for a series of behavioral approaches (see below), there are some common factors. In both groups, therapists worked on the sense of loneliness by promoting mutual support between the couples and continuous dialogues among them. Another common factor between the two interventions was the weekly assignment of tasks to families, which allowed an increase in participants’ motivational level. At the end of each meeting, therapists assigned homework, which was discussed at the next meeting. The weekly commitment, linked to the condition of having a goal to be completed in a short time and the ability to share their experiences in a group context, motivated participants to actively engage in the assigned tasks. However, there are differences concerning the content of the exercises. The group enrolled for the ACT matrix protocol focused its work mainly on increasing the level of psychological flexibility, paying particular attention to the role of values and committed action, whereas the PT group underwent a protocol that applied the classic behavioral tasks related to the management of one’s child.

### 2.5. ACT Matrix

The matrix [35] is an ACT protocol that is usually presented visually and consists of two intersecting lines that make up four quadrants, which provide a “point of view” on one’s psychological actions and experiences. The vertical line is the line of experience, the upper part corresponds to the experience of life linked to the five senses—sight, hearing, taste, smell, and touch—whereas the lower part refers to internal experiences, such as thoughts and feelings (internal/mental experience). The horizontal line is the behavior line, wherein the left side concerns the actions that perform the function of moving us away from experiences, emotions, and unwanted thoughts (experiential avoidance), and the right side indicates the actions we take to get closer to our values (committed action). The current study was designed to produce preliminary data on the effectiveness of the use of the matrix in two groups of parents of children with ASD to evaluate how improvement in terms of psychological flexibility can affect perceived parental stress levels. Furthermore, the correlation between parental psychological well-being and the intensity of children’s problem behaviors was evaluated, also assessing their emotional competence by hypothesizing an improvement in terms of identification and discrimination of emotions. The ACT protocol group homework exercises were specifically designed to improve the parents’ psychological well-being.

### 2.6. PT

PT is a training program for parents to teach behavioral management skills for children. The purpose of a PT program is to promote positive parenting and reduce the behavioral risks of children [36]. Numerous studies have examined the effectiveness of PT interventions to improve health outcomes for children with ASD [17,37,38]. Research interventions have demonstrated the efficacy of these interventions to improve the behavioral performance of children, increase parents’ ability to manage children, and decrease the presence of problem behaviors [17]. Such studies highlight improvements in parental competence in managing children in behavioral terms but do not examine changes in parental psychological well-being. Many studies show that coping strategies, social support, and parenting effectiveness are psychological factors that affect parental stress levels [39]. PT interventions conducted in groups offer an effective solution for modifying parental behavior by providing social support and new coping strategies. The PT protocol group received behavioral tasks related to child management.

### 2.7. Outcome Measures

We collected primary and secondary outcome measures. A pre-/post-treatment assessment was carried out relating to the measurement and change of the parents’ psychological flexibility during the intervention. The primary outcome measures used were the Acceptance and Action Questionnaire II (AAQ-II) [40], to measure the psychological flexibility of the person and their ability to stay in touch with their emotions, and the Home Situation Questionnaire (HSQ-ASD), which gives objective measures of the perception and influence of children’s behavior in their parent’s life. Secondary outcome measures were the Valued Living Questionnaire (VLQ), which allows for identifying the important areas for the person, the Mindfulness Attention Awareness Scale (MAAS), which measures the tendency of an individual to intentional awareness, and the Parental Stress Index (PSI) to assess the stress level pre- and post-treatment.

### 2.8. Primary Outcome Measures

#### 2.8.1. Acceptance and Action Questionnaire (AAQ-II)

The AAQ-II [40] is a ten-item test with answers on a scale from 1 (never true) to 7 (always true) to measure the person’s psychological flexibility and their ability to stay in touch with their emotions. The items focus on the willingness to separate unwanted private events, on the ability to live in the present moment, and on the commitment to adopt flexible and valuable actions during the experience of internal negative events. The higher the score, the greater the psychological flexibility.

#### 2.8.2. Home Situation Questionnaire (HSQ-ASD)

The HSQ-ASD [41] is a caregiver-rated scale designed to assess the severity of disruptive and non-compliant behaviors in children. The scores obtained with this scale refer to the parent’s perception of their child’s behavioral manifestations. Within the scale, data are collected on inflexibility and avoidance manifested by the child. This modified and revised version for ASD consists of 27 elements. Parents are asked to indicate if their children have problems with compliance in these situations and, if so, to rate severity on a Likert scale of 0 to 9, with higher scores indicating greater non-compliance.

### 2.9. Secondary Outcome Measures

#### 2.9.1. Valued Living Questionnaire (VLQ)

The VLQ [42] is a questionnaire exploring some areas of life that people consider important such as family relationships, marriage/being in a couple, intimate relationships, friends, social relationships, work, culture/training, leisure/entertainment, spirituality, civic commitment/community life, and self-care. The questionnaire provides an importance score, in which, for each dimension, the person is asked to rate on a scale from 1 (not at all important) to 10 (extremely important) how important that area of their life is. They are then asked to evaluate how much, in the last week, their actions have been congruent with the values considered, on a scale ranging from 1 (completely incongruent actions) to 10 (completely congruent actions). The questionnaire provides a composite score that is calculated through the application of a formula from which the degree to which the person, in relation to the importance given to a specific area, behaves in a manner consistent with it being obtained. The indices obtained from this scale are the VLQI, which detect how much importance the subject assigns to the ten measured domains, the VLQC which measures how much the person is engaged in behaviors that are congruent with their values, and the VLQtot which indicates the composite score with which the degree to which the person is in contact with their personal values in everyday life is measured.

#### 2.9.2. Mindful Attention Awareness Scale (MAAS)

The Mindful Attention Awareness Scale (MAAS) [43] measures individual differences in daily awareness states. The 15 items of the MAAS are preceded by an introductory statement: “Below you will find some statements about your daily experiences. Using the proposed scale from 1 to 7, indicate the frequency in which each of the experiences described has happened to you. Please base your answers on your actual experiences, rather than thinking about what the experiences should be”. Then, respondents rate the 15 elements of the scale on a 7-point Likert-type scale, from 1 (almost always) to 7 (almost never). Higher values indicate higher levels of awareness.

#### 2.9.3. Parental Stress Index/Short Form (PSI/SF)

The PSI/SF [44] is a self-assessment questionnaire. The administration and compilation of the test take about 10–15 min. The Italian validation of the test is based on the short form which derives from the extended form. The hypothesis underlying the test is that parental stress levels are given by the interaction of 3 different factors: 1. characteristics of the children, 2. characteristics of the parent, and 3. aspects related to the parental situation. The short form is composed of 36 items, divided into three subscales: (1) Parental Distress (PD), which taps into parental feelings; (2) Parent–Child Dysfunctional Interaction (P–CDI), which focuses on the perception of the child as not responding to parental expectations; and (3) Difficult Child (DC), which is centered on some of the characteristics of the child that make it easy or difficult to manage them.

### 2.10. Statistical Analysis

Considering the small sample size, non-parametric statistics (Mann–Whitney U-tests and the Wilcoxon signed-rank test) were applied in order to analyze the effects of group and intervention. We adjusted the alpha level using a Šidák correction for hierarchical multiple comparisons, alpha = 0.025 for the primary measures, total AAQ-II and HSQ-ASD scores, alpha = 0.01 for the secondary measures, total PSI/SF, MAAS and VLQ scores; alpha = 0.025 for the two VLQ subscales; alpha = 0.025 for the two VLQ subscales; alpha = 0.025 for the two HSQ-ASD subscales; and alpha = 0.017 for the three PSI/SF subscales. Therefore, with alpha = 0.025 and power = 0.8, the projected sample size needed for large effect size d = 1.0 (GPower 3.1) is approximately N = 36 for this simplest pre- and post-intervention comparison. Thus, our proposed sample size of 40 should be adequate for the main objective of this study.

## 3. Results

### 3.1. Clinical Enrollment

We enrolled N = 66 pairs of parents of 66 children with ASD, aged 30 months to 13 years. A first screening based on inclusion criteria was implemented and *n* = 20 pairs of parents of children with ASD (18 male/2 female) fully met the admission criteria and were enrolled in the present study. Parents were randomly assigned to either the matrix group (TG) or the PT group (CG). Groups were pair matched according to the children’s DQ and parental education. Twenty pairs of parents completed all phases of the rehabilitation protocols and were included in the statistical analysis (Figure 2).

In the TG, the average child’s age was 6.9 years, DQ was 88.6, and parental age was 40.6 with 16.0 years of formal education. In the CG, the average child’s age was 5.8 years, DQ was 86.1, parental age was 42.0 with 15.8 years of formal education. No significant differences were detected in any demographic variables (Table 1). All parents, except one who only had a middle school diploma, had a high school or college degree; 18 parents were office workers, 6 were employed professionals, 6 were freelance professionals, 4 were hard workers/artisans, and 6 (all females) were stay-at-home parents with their children. All children of included parents were Italian and attended mainstream public schools for 27 h a week with a special teacher for 10–12 h. Furthermore, they received between 2 and 4 h of intervention outside our facility, consisting of a mix of speech and language therapy, occupational therapy, and behavioral intervention. None of the parents received other forms of parent training or psychological support.

### 3.2. ACT Versus PT Treatments

The baseline measures of outcome variables showed no significant differences between groups (Table 1). The only marginal difference was on the HSQ-ASD total score, M = 3.65 for TG and M = 2.74 for CG, with U = 129, W = 339, Z = −1.90, *p* = 0.056.

Comparing the TG and the CG on intervention gains (Table 2), a main effect of group was found for both the AAQ-II total score, U = 0.00, W = 210, Z = −5.51, *p* < *0*.001, and the HSQ-ASD total score, U = 302, W = 512, Z = 2.78, *p* = 0.005, with the TG showing a significantly higher degree of improvement compared to the CG. Similarly, the differences were also significant for secondary measures and all subscales, with the exception of PSI/SF subscales that were marginally significant on PD and CDI but showed no difference in DC.

Furthermore, after the intervention, we found a significant improvement in TG on AAQ-II, U = 400, W = 610, Z = 5.42, *p* < 0.001 (Figure 3; Table 2), with a gain of 64% in the total score, and all parents in the group improved between 16 and 32 points (Table 3). Conversely, the CG did not improve, with a maximum range of change between −2 and 4 points, U = 221, W = 431, Z = 0.577, *p* = 0.583. A smaller improvement, and only significant before correction for multiple comparisons, was also present for HSQ-ASD in TG (Figure 3), U = 119, W = 329, Z = −2.18, *p* = 0.028, with a reduction of 32% in the total score, while for CG there was no change, U = 208, W = 428, Z = 0.488, *p* = 0.383. A similar pattern held for VLQ, MAAS, and marginally for PSI/SF (Table 3).

## 4. Discussion

In this study, we demonstrated that after six months of training using the ACT matrix protocol, ASD parents showed evident improvement in all primary and secondary outcome measures with respect to another group of demographically and clinically matched ASD parents who underwent PT treatment. Indeed, after treatment, we detected significantly increased scores in the psychological flexibility (AAQ), awareness states (MAAS), personal values in everyday life (VLQ), and parental stress (PSF), whereas reduced scores were detected in the parents’ perception of their child’s disruptive behaviors (HSQ-ASD).

Taking care of ASD parents can be especially demanding because physical (e.g., dressing and bathing) and cognitive impairments (e.g., preparing foods or studying) often compromise daily activities. Parents may perceive a burden due to the complexity of caring, especially if they feel that they have been left alone without healthcare workers’ support. For this reason, it is mandatory to develop new behavioral training strategies for helping ASD parents. This study highlights how intervention based on the ACT principles allows us to get a great improvement in terms of psychological well-being and a decrease in stress levels. ACT treatment has already been used in other ASD studies demonstrating its effectiveness in reducing depressive symptoms in parents of children with ASD [45]. Others suggest that the ACT intervention improves the perception of parental competence by favoring the vision and perception of feeling like a more aware and competent parent in the management and understanding of their autistic child [46]. However, this is the first study where the evidence of ACT-related beneficial effects was obtained using a randomized controlled study.

Another additional new finding provided by our study is the presence of a reduced perception of disruptive behaviors by the parents. It is well known that parental competence and problem-solving skills are influenced by the disruptive behavior of their kids [47,48]. Disruptive behaviors, including noncompliance, oppositional behaviors, aggression, and self-injury, affect as many as 50% of children with ASD. These types of behaviors contribute to parental stress and strain [49,50] and may amplify the caregiver’s burden [51]. However, research suggests that child-specific factors such as age, intellectual functioning, and behavioral disorders appear to influence parental perspectives [52]. Our findings suggest that, after training with the ACT matrix, parents may acquire new strategies or psychological flexibility, which is useful to induce a cognitive reconstruction of their life and emotional reactions to stress.

It is important to bear in mind that our significant findings were obtained comparing parental psychological statuses with respect to demographically matched individuals undergoing another well-known behavioral treatment, parental training. The primary goal of PT is to provide parents with specific techniques to manage children’s behavioral problems. Indirectly, group-based PT is able to provide social support and decreased parental stress [31,32,33,34]. Otherwise, ACT is a contextual cognitive behavioral therapy that emphasizes processes of acceptance, awareness, and value-based strategies to address a wide range of psychological and behavioral health problems [6]. The behavioral intervention focused on improving psychological flexibility allows parents to provide internal resources to cope with problematic situations related to their child’s diagnosis of autism [45,53]. However, behavior intervention alone may not be enough to prevent the manifestation of anxious-depressive symptoms in the parents. As suggested by Jones et al. [54], to face this kind of neurodevelopmental disorder, emotional support to the couple may become the most important protective factor for maintaining adequate levels of psychological well-being.

### Limitations

There are some limitations to this study. First, we should acknowledge that the sample size was small, which may have accounted for the lack of significant improvement shown by the experimental group in our non-parametric analysis. Nevertheless, the ancillary analysis using repeated measures ANOVA and bias-corrected accelerated bootstrap suggested a small but significant effect also for the experimental group. Second, it is worth noting that we did not find any significant improvement in the perception of the disruptive and non-compliant behavior of the children in the control group. This result may be dependent on the characteristics of the group because a group-based behavioral approach might not be useful for all parents with respect to a behavioral approach tailored to the individual needs [55]. Furthermore, the control group protocol was based only on managing the behavior of the children, so it is possible that the parents did not achieve a change in the HSQ scores because they were not offered any tools to change their point of view about the diagnosis and functioning of their child.

## 5. Conclusions

Our study supports the idea that an intervention based on the group sharing of experiences may be more effective than a treatment based on individual support (PT) to modify and improve the perception of the severity of one’s child’s behavioral symptoms. Moving from changes in parents’ perception of their child, the ACT protocol matrix can produce a better and significant improvement in parental stress levels. We believe that the positive effects should be related to the focusing of behavioral treatment on psychological flexibility, acceptance process, and contact with values. ACT-oriented treatment involving parents of children with autism can produce greater improvements in relation to parental stress than behavioral treatment aimed only at children and can produce significant behavioral activation of the parents towards values. These results indicate that an ACT-oriented intervention can be considered among the possible interventions to be proposed to parents of children with autism to support psychological well-being by intervening on the whole family. Finally, here we provided compelling new evidence that the ACT protocols should be included in daily practice to support the development of new parenting skills in ASD parents.

## Figures and Tables

**Figure 1 brainsci-11-00880-f001:**
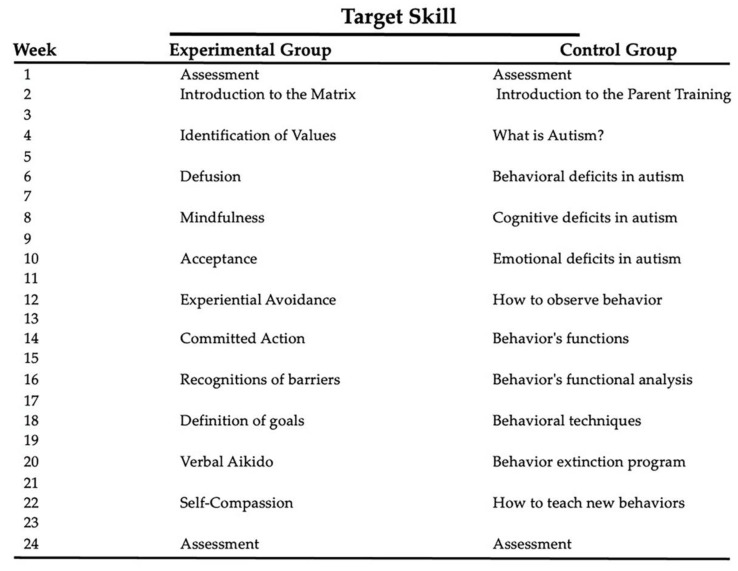
ACT matrix meetings and Parental Training meetings.

**Figure 2 brainsci-11-00880-f002:**
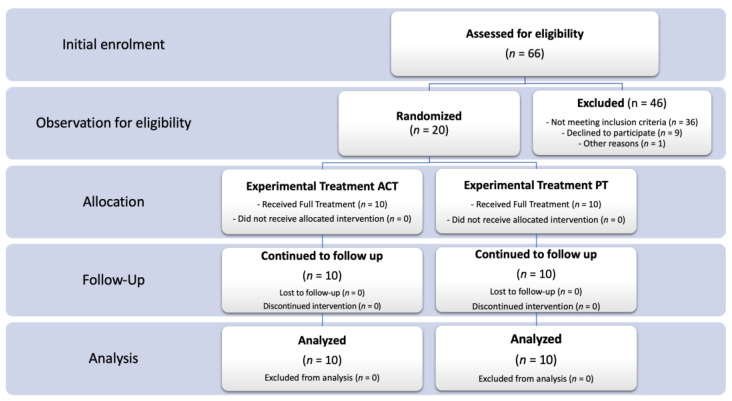
CONSORT Flow diagram showing the phases of a parallel randomized trial of two groups of ASD parent pairs underwent experimental (ACT: Acceptance and Commitment Therapy) or conventional (PT: Parent Training) training interventions.

**Figure 3 brainsci-11-00880-f003:**
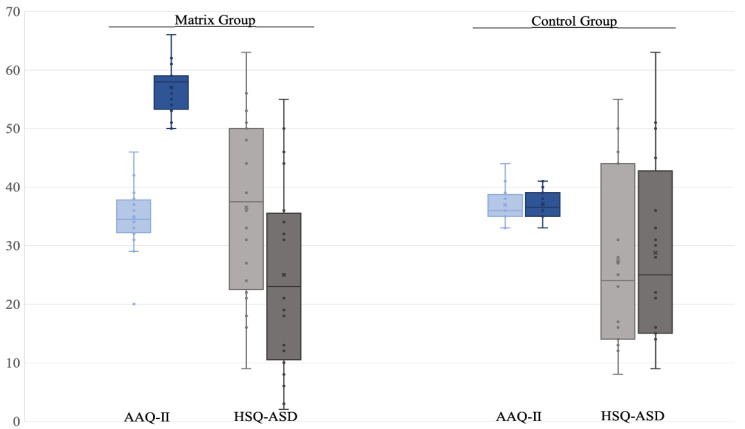
Pre- and post-intervention comparison between and within groups on primary outcome measures. Pre-intervention: in each block, on the left, drawn with lighter colors. Post-intervention: in each block, on the right, drawn with darker colors.

**Table 1 brainsci-11-00880-t001:** Demographic characteristics and pre-intervention outcome variables of the sample.

Variable	Matrix Group (*n* = 20)				PT Group (*n* = 20)				Comparison between Groups			
	Mean	SD	Mdn	Full Range	Mean	SD	Mdn	Full Range	U	W	Z	*p-*Value
Child Age *	6.90	1.66	6.50	5–10	5.80	2.39	5.00	4–10	28.0	83.0	−1.69	0.105
Child DQ *^,$^	88.6	11.6	91.0	72.4–111	86.1	10.6	85.4	70.0–107	43.0	98.0	−0.529	0.631
Parental Age	40.6	5.34	40.5	31–49	42.0	5.71	41.0	37–58	182	392	−0.489	0.640
Parental Education ^$^	16.0	2.99	18.0	8–18	15.8	2.55	18.0	13–18	215	425	0.490	0.678
AAQ-II	34.6	5.35	34.5	20–46	36.9	3.06	36.0	33–44	257	467	1.56	0.121
HSQ-ASD	4.12	1.50	3.80	1.4–6.9	3.29	1.68	2.90	1.1–6.7	129	339	−1.90	0.056
VLQ	84.4	7.14	87.0	69–91	83.4	7.55	85.0	64–91	181	391	−0.520	0.620
MAAS	42.1	4.63	41.5	35–55	40.0	3.22	39.5	35–48	139	349	−1.65	0.102
PSI/SF	30.8	10.8	31.0	12–55	29.4	8.87	27.0	15–43	149	359	−1.38	0.174

DQ: Developmental Quotient, AAQ-II: Acceptance and Action Questionnaire II, HSQ-ASD: Home Situation Questionnaire, VLQ: Valued Living Questionnaire, MAAS: Mindfulness Attention Awareness Scale, PSI: Parental Stress Index, Mdn: Median, U: Mann–Whitney U, W: Wilcoxon W, SD: Standard Deviation. * Children statistics are based on a sample of 10 children for each group. ^$^ Used for pair matching in randomization.

**Table 2 brainsci-11-00880-t002:** Delta of improvements in outcome measures between matrix and Parent Training groups.

		Matrix Group (*n* = 20)								Control Group (*n* = 20)								Intervention Outcome Comparison between Groups			
		Baseline				Post- Pre- Difference				Baseline				Post- Pre- Difference							
Test	Factor	Mean	SD	Mdn	Full Range	Mean	SD	Mdn	Full Range	Mean	SD	Mdn	Full Range	Mean	SD	Mdn	Full Range	U	W	Z	*p-*Value
AAQ-II	*Total*	34.6	5.35	34.5	20–46	22.3	4.77	21.5	16.0–32.0	36.9	3.06	36.0	33–44	.250	1.25	0.000	−4.00–2.00	.000	210	−5.51	<0.001 *
HSQ-ASD	SI	4.12	1.50	3.80	1.4–6.9	−1.18	1.64	−0.950	−4.40–1.90	3.29	1.68	2.90	1.1–6.7	−0.020	0.900	0.300	−2.30–1.00	302	512	2.78	0.005 *
	DA	3.37	1.84	3.55	0.5–7.2	−1.18	1.49	−0.650	−4.60–0.50	2.42	1.68	1.55	0.5–5.8	0.320	0.640	0.400	−1.20–1.40	345	555	3.92	<0.001 *
	*Total*	3.65	1.52	3.75	0.9–6.3	−1.16	1.41	−0.700	−4.50–0.80	2.74	1.49	2.40	0.8–5.5	0.140	0.590	0.200	−1.30–1.00	337	547	3.72	<0.001 *
VLQ	I	84.4	7.14	87.0	69–91	3.00	4.23	2.00	−7.00–14.0	83.4	7.55	85.0	*64–91*	0.250	5.38	0.000	−8.00–20.0	78.5	288	−3.37	0.001 *
	C	47.8	3.68	47.5	42–55	22.1	6.27	22.5	9.00–35.0	38.9	5.14	47.5	42–60	−0.150	1.66	0.000	−4.00–3.00	0.000	210	−5.51	<0.001 *
	*Total*	40.1	4.16	40.4	34–49	21.3	6.79	20.95	4.20–38.0	41.8	3.68	42.0	35–49	−0.100	1.30	0.000	−2.70–2.70	0.000	210	−5.45	<0.001 *
MAAS	*Total*	42.1	4.63	41.5	35–55	22.8	4.38	22.5	13.0–31.0	40.0	3.22	39.5	35–48	0.150	1.53	0.000	−4.00–3.00	0.000	210	−0.545	<0.001 *
PSI/SF	PD	30.8	10.8	31.0	12–55	−2.55	9.34	−4.00	−20.0–19.0	29.4	8.87	27.0	15–43	2.30	8.26	0.000	−15.0–20.0	277	487	2.08	0.038 ^$^
	CDI	29.8	7.42	28.0	16–49	−4.35	5.64	−3.00	−15.0–5.00	27.8	8.10	26.5	16–44	−0.600	7.82	−0.500	−21.0–13.0	269	479	1.88	0.063
	DC	37.5	7.94	36.0	25–53	0.000	12.5	−0.500	−24.0–26.0	38.8	7.06	37.5	26–52	0.050	7.77	1.00	−17.0–12.0	213	423	0.352	0.738
	*Total*	98.1	11.9	97.5	80–132	−6.90	6.12	−6.00	−19.0–3.00	92.9	11.1	90.0	77–113	1.80	7.19	1.50	−10.0–17.0	328	538	3.47	<0.001 *

Mdn: Median, U: Mann–Whitney U, W: Wilcoxon W, SD: Standard Deviation. ^$^ no longer significant after controlling for multiple comparisons. AAQ-II: Acceptance and Action Questionnaire II. HSQ-ASD: Home Situation Questionnaire. VLQ: Valued Living Questionnaire. VLQI: VLQ Importance. VLQC: VLQ Congruent. MAAS: Mindfulness Attention Awareness Scale. PSI: Parental Stress Index. * significant after correction for multiple comparisons.

**Table 3 brainsci-11-00880-t003:** Pre- and post-intervention comparisons in outcome measures for matrix and Parent Training groups.

		Matrix Group (*n* = 20)				Parent Training Group (*n* = 20)			
Test	Factor	U	W	Z	*p-*Value	U	W	Z	*p-*Value
AAQ-II	*Total*	400	610	5.42	<0.001 *	221	431	0.577	0.583
HSQ-ASD	SI	124	334	−2.06	0.040	208	418	0.217	0.841
	DA	122	332	−2.10	0.035	232	442	0.881	0.383
	*Total*	119	329	−2.18	0.028 ^$^	218	428	0.488	0.640
VLQ	I	260	470	1.64	0.102	186	396	−0.369	0.718
	C	400	610	5.42	<0.001 *	198	408	0.055	0.968
	*Total*	398	608	5.36	<0.001 *	199	409	0.014	0.989
MAAS	*Total*	400	610	5.42	<0.001 *	204	414	0.110	0.925
PSI/SF	PD	173	383	−0.718	0.478	229	439	0.787	0.445
	CDI	135	345	−1.78	0.081	198	408	−0.041	0.968
	DC	195	405	−0.136	0.904	199	409	0.027	0.989
	*Total*	126	336	−2.01	0.046 ^$^	226	436	0.718	0.478

U: Mann–Whitney U, W: Wilcoxon W, SD: Standard Deviation. ^$^ no longer significant after controlling for multiple comparisons. * significant after correction for multiple comparisons. the same explanation with Table 2.

## Data Availability

The data presented in this study are available on request from the corresponding author.

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
