# Peer review of "The Effect of Acceptance and Commitment Therapy for Improving Psychological Well-Being in Parents of Individuals with Autism Spectrum Disorders: A Randomized Controlled Trial"

_brainsci, 2021, doi:10.3390/brainsci11070880_

Round 1

Reviewer 1 Report

Dear Authors,
I read your article carefully. He is interested and I recommend a series of changes:

  • the abstract is unclear and does not highlight the most important aspects of the article. Please modify the website to highlight the most important segments of the article;
    - line 47-51 - please rewrite, it is not very clear what you want to say;
    - I suggest you convert table 1 into a figure ... it's much easier to understand ... and I think, more impactful;
    - discussions are too short, please enter new discussions based especially on your results;
    - there are a series of typos in English - please correct;

Thank you

BR,

Author Response

Dear Editor

please find in the following our point-by-point response: 

I read your article carefully. He is interested and I recommend a series of changes:

  • the abstract is unclear and does not highlight the most important aspects of the article.
    • Following the reviewer’s suggestion the abstract has been modified.
  • Please modify the website to highlight the most important segments of the article;
    • We don’t understand the meaning of term “website”. If the reviewer means “structured abstract”, even if it is not required by Brain Sciences journal, we modified the abstract accordingly.
  • line 47-51 - please rewrite, it is not very clear what you want to say
    • The paragraph was rewritten.
  • I suggest you convert table 1 into a figure ... it's much easier to understand ... and I think, more impactful;
    • Following the reviewer’s suggestion, the table 1 has been converted to a Figure.
  • discussions are too short, please enter new discussions based especially on your results;
    • In the discussion we reported the main findings of this paper that could be of interest for further advances in this field of study. We do not believe that any additional point should be discussed. We kindly ask the reviewer to formulate a more specific question about the issue that would not have been highlighted in our paper.
  • there are a series of typos in English - please correct;
    • Done

Reviewer 2 Report

I thank the editor for the possibility to revise the paper entitled “The effect of Acceptance and Commitment Therapy for improving psychological well-being in parents of individuals with autism spectrum disorders: a randomized controlled trial”. The study purpose was to evaluate the efficacy of the therapy protocol comparing the Parent Training (PT) program to a conventional therapy on parents of children with ASD.

Although the study was interesting, I think the paper should be revised.

Following the minor revisions suggested.

In general, the introduction was well organized and readable. Furthermore, the HP was clear.

  • Line 70: authors discussed the importance of the intervention focused on parental acceptance of child’s diagnosis, I think it was a pivotal point and I suggest authors to improve the references reading, for example:
    • Lecciso, F., Petrocchi, S., Savazzi, F., Marchetti, A., Nobile, M., & Molteni, M. (2013). The association between maternal resolution of the diagnosis of autism, maternal mental representations of the relationship with the child, and children’s attachment. Life Span Disabil16, 21-38.
    • Russell, G., & Norwich, B. (2012). Dilemmas, diagnosis and de-stigmatization: Parental perspectives on the diagnosis of autism spectrum disorders. Clinical child psychology and psychiatry17(2), 229-245.
  • Regarding the intervention, I do not understand whether the children were paired or not. I understand that parent were randomly included in experimental or control group, but I do not understand whether the children were similar between the two group. They had similar deficits or not?
  • Because the small sample size and because data did not met criteria for parametric analyses, why authors did not carry out non parametric analysis for longitudinal data? Why they prefer MANOVA?
  • Regarding the results and discussion I have not suggestion. They are clear and well structured.

Author Response

Dear Editor

please find in the following our point-to-point response:

In general, the introduction was well organized and readable. Furthermore, the HP was clear.

  • Line 70: authors discussed the importance of the intervention focused on parental acceptance of child’s diagnosis, I think it was a pivotal point and I suggest authors to improve the references reading, for example:
      • Lecciso, F., Petrocchi, S., Savazzi, F., Marchetti, A., Nobile, M., & Molteni, M. (2013). The association between maternal resolution of the diagnosis of autism, maternal mental representations of the relationship with the child, and children’s attachment. Life Span Disabil16, 21-38.
      • Russell, G., & Norwich, B. (2012). Dilemmas, diagnosis and de-stigmatization: Parental perspectives on the diagnosis of autism spectrum disorders. Clinical child psychology and psychiatry17(2), 229-245.

    • Re: Done

  • Regarding the intervention, I do not understand whether the children were paired or not. I understand that parent were randomly included in experimental or control group, but I do not understand whether the children were similar between the two group. They had similar deficits or not?

    • Re: Children were paired according to the following demographic variables: developmental quotient, parental education. As showed in table 1 there are no differences in demographic variables between groups neither in ASD children nor in parents, also for the other variables (child age and parental age), descriptions were added in methods and result sections. To further examine the group differences, we now reported also the initial value of outcome variables for each group.

  • Because the small sample size and because data did not met criteria for parametric analyses, why authors did not carry out non parametric analysis for longitudinal data? Why they prefer MANOVA?

    • Re: We thank the reviewer for noticing, it was actually a versioning error, we considered from the beginning the use of non-parametric analyses, as reported in the method section and used in the result section. In the beginning we planned to add also a parametric analysis as supplementary materials but we didn’t do that in the last version. Therefore, we removed the description from the methods accordingly.

  • Regarding the results and discussion I have not suggestion. They are clear and well structured.

    • Re: We would like to thank the reviewer for kind suggestions and appreciations.

Reviewer 3 Report

I'm very impressed from encouraging results obtained by the authors. 
However, I think that an encouraging result it must not make us forget that this study has an enormous limitation: the small number of enrolled subjects. The authors are aware of this limit and they brought it in the Limits.
I suggest that the authors be more cautious.
In the abstract, they report:
'This randomized controlled trial highlighted the importance to include ACT protocols in daily practice to support the development of new parenting skills'.
This sentence is too pretentious. 
I suggest the following:
'ThE RESULTS OF THIS randomized controlled trial, IF REPEATED WITH A LARGE NUMBER OF SUBJECTS, COULD OPEN THE WAY to include ACT protocols in daily practice to support the development of new parenting skills'.
At line 433 'Institutional Review Board Statement' you have to complete it.
Methods are well described. Statystical analysis are well structured (the limit is the sample).
Introduction and Discussionare compelling.

Author Response

Dear Editor

please find in the following our point-by-point response:

  • In the abstract, they report: 'This randomized controlled trial highlighted the importance to include ACT protocols in daily practice to support the development of new parenting skills'. This sentence is too pretentious. I suggest the following: 'ThE RESULTS OF THIS randomized controlled trial, IF REPEATED WITH A LARGE NUMBER OF SUBJECTS, COULD OPEN THE WAY to include ACT protocols in daily practice to support the development of new parenting skills'.

    • Re: We would like to thank the reviewer for this kind suggestion. Abstract has been modified accordingly.

  • At line 433 'Institutional Review Board Statement' you have to complete it.

    • Re: Done

  • Methods are well described. Statystical analysis are well structured (the limit is the sample). Introduction and Discussionare compelling.

    • Re: We would like to thank the reviewer